# Binding of HMGN proteins to cell specific enhancers stabilizes cell identity

Bing He[1], Tao Deng[1], Iris Zhu[2], Takashi Furusawa[1], Shaofei Zhang[1], Wei Tang[3], Yuri Postnikov[1], Stefan Ambs [3], Caiyi Cherry Li[4], Ferenc Livak[4], David Landsman [2] & Michael Bustin[1]

The dynamic nature of the chromatin epigenetic landscape plays a key role in the establishment and maintenance of cell identity, yet the factors that affect the dynamics of the epigenome are not fully known. Here we find that the ubiquitous nucleosome binding proteins HMGN1 and HMGN2 preferentially colocalize with epigenetic marks of active chromatin, and with cell-type specific enhancers. Loss of HMGNs enhances the rate of OSKM induced reprogramming of mouse embryonic fibroblasts (MEFs) into induced pluripotent stem cells (iPSCs), and the ASCL1 induced conversion of fibroblast into neurons. During transcription factor induced reprogramming to pluripotency, loss of HMGNs accelerates the erasure of the MEF-specific epigenetic landscape and the establishment of an iPSCs-specific chromatin landscape, without affecting the pluripotency potential and the differentiation potential of the reprogrammed cells. Thus, HMGN proteins modulate the plasticity of the chromatin epigenetic landscape thereby stabilizing, rather than determining cell identity.

[1] Protein Section, Laboratory of Metabolism, Center for Cancer Research, National Cancer Institute, National Institutes of Health, Bethesda, MD 20892, USA. [2] Computational Biology Branch, National Center for Biotechnology Information, National Library of Medicine, National Institutes of Health, Bethesda, MD 20892, USA. [3] Laboratory of Human Carcinogenesis, Center for Cancer Research, National Institutes of Health, Bethesda, MD 20892, USA. [4] Laboratory of Genomic Integrity, Center for Cancer Research National Cancer Institute National Institutes of Health, Bethesda, MD 20892, USA. These authors contributed equally: Tao Deng, Iris Zhu. Correspondence and requests for materials should be addressed to M.B. (email: bustinm@mail.nih.gov)

Proper maintenance of cell identity, a requirement for correct differentiation and for preventing disease, is crucially dependent on the dynamic nature of the epigenetic landscape encoded in chromatin. Preprogrammed changes in cell fate occurring during differentiation or in response to biological stimuli, are invariably associated with significant changes in the epigenetic landscape, most notably at tissue-specific enhancer regions[1,2]. While programmed chromatin remodeling is an integral part of development and a requirement for mounting proper biological responses, unprogrammed epigenetic changes can destabilize the maintenance of cell identity leading to diseases[3,4]. Thus, the epigenetic landscape needs to be sufficiently stable to prevent deleterious changes in cell identity, yet sufficiently permissive to allow adequate responses to preprogrammed events leading to advantageous changes in cell identity.

Changes in the epigenetic landscape are also seen during ectopic transcription factor induced reprogramming of mature cells to pluripotency and during direct cell lineage fate conversion[5–7]. The ectopically expressed transcription factors are the main drivers of the epigenetic changes that lead to changes in cell identity; however, factors that regulate chromatin topology, nucleosome organization, histone modifications and enhancer accessibility seem to affect the efficiency of cell reprogramming[8–11]. For example, the ubiquitous linker H1 protein family, a major global regulator of chromatin structure and function, undergoes significant compositional changes during reprogramming and seems to play important roles in mediating the establishment of cell identity[12–14]. Likely, additional global regulators of chromatin organization, such as the chromatin binding High Mobility Group (HMG) architectural proteins[15], could play a role in safeguarding cell identity[16,17], however this possibility has not yet been fully explored. Chromatin architectural proteins such as H1 and HMGs are ubiquitously expressed in the nuclei of all vertebrate cells potentially affecting epigenetic processes and the maintenance of cell identity in many cell types.

Here we examine the possibility that the high mobility group N (HMGN) proteins act as chromatin modulators that affect epigenetic plasticity, i.e. the ability to alter the epigenetic landscape, and play a role in maintaining cell identity. The ubiquitous HMGNs bind dynamically to nucleosomes, the building block of the chromatin fiber, without DNA sequence specificity[18]. The interaction of HMGN proteins with nucleosomes promotes chromatin decompaction because it reduces the chromatin binding of H1[19,20] and obstructs access to the nucleosome acidic patch[21]. Although HMGNs bind to chromatin without DNA sequence specificity, genome-wide analysis in mouse embryonic fibroblasts (MEFs) suggests that they tend to colocalize with DNA hypersensitive sites (DHS) and fine-tune enhancer organization[22,23].

We now analyze the genome-wide organization of HMGNs in the chromatin of several cells types and find that these proteins colocalize with epigenetic marks of active chromatin and with cell-specific regulatory sites, raising the possibility that they play a role in cell fate decisions. To test this possibility, we study the conversion of wild type and $Hmgn1^{-/-}/Hmgn2^{-/-}$ (DKO) MEFs, which lack both HMGN1 and HMGN2, into pluripotent cells following the ectopic expression of Oct4, Sox2, Klf4, and c-Myc (OSKM). During reprogramming to pluripotency, these exogenous transcription factors were shown to first target somatic enhancers to initiate the gradual erasure of the MEF-specific epigenetic landscape and the gradual establishment of an embryonic stem cell (ESC) specific epigenetic landscape[6,7,24]. We now find that loss of HMGNs enhances the rate of these epigenetic changes and improves the reprogramming efficiency of MEFs into pluripotent cells. Likewise, we find that loss of HMGN accelerates the direct conversion of MEFs into neuronal cells. Our studies reveal an essential role for the ubiquitous HMGN proteins in regulating chromatin dynamics during reprogramming, and indicate that their presence at cell-type specific regulatory sites stabilizes the cellular epigenetic landscape that maintains cell identity.

## Results

**HMGNs localize to lineage specific regulatory sites.** Chromatin immunoprecipitation (ChIP) analysis in mouse embryonic fibroblasts (MEFs), embryonic stem cells (ESCs) and resting B cells (rBs) reveals that the location of the two major HMGN variants, HMGN1 and HMGN2, coincides throughout the genome of these cells (Supplementary Fig. 1b). To determine the organization of HMGNs within the context of the cellular epigenetic landscape, we performed ChIP analyses of several histone modifications that mark either transcriptionally active or transcriptionally silent chromatin, the DNase I hypersensitive sites (DHS) in the 3 types of cells, and the Assay of Transposase Accessible Chromatin (ATAC) site accessibility in MEFs. Comparison of the genome wide organization of HMGNs with data generated in our laboratory and with data available in Encode[6,25], indicates that in all the three cell types, HMGNs colocalize with epigenetic marks of active chromatin such as H3K4me3, H3K27ac and H3K4me1, and with DNase I or ATAC sensitive sites, that mark chromatin regions which are relatively de-condensed and preferentially accessible to regulatory factors. In contrast, HMGNs are relatively depleted from genomic sites enriched in histone marks of condensed, transcriptionally silent chromatin regions such as H3K27me3 or H3K9me3 (Fig. 1a and Supplementary Fig. 1a).

Among the three cell types, the location of HMGN1 and HMGN2 variants overlaps at annotated promoter regions but shows little overlap at non-promoter regions (Fig. 1b). In each cell type, the subset of genes that are expressed with high cell specificity (Supplementary Fig. 1d) shows high HMGN1 and HMGN2 occupancy at the promoters (Fig. 1c and Supplementary Fig. 1e), suggesting that the genome wide organization of HMGNs is cell-type specific. Furthermore, when the genes within a cell type are ranked by expression levels (Fig. 1d), we find a direct correlation between gene expression levels and HMGN occupancy at transcription start sites: in each cell type, HMGN1 and HMGN2 occupancy is highest at the most highly expressed genes and lowest at the least expressed genes (Fig. 1e and Supplementary Fig. 1f). Thus, at annotated promoters, the HMGN occupancy levels are directly related to the cell-type specific gene expression levels.

In agreement, chromatin regions which show cell-type specific enrichment in H3K27ac, H3K4me1 and H3K4me3 also shows a cell-type specific pattern in the location of HMGN1 and HMGN2 (Fig. 1f and Supplementary Fig. 2a). For example, genomic regions that contain these histone modifications in ESCs but not in MEFs or rBs, show HMGN occupancy only in ESCs. Likewise, in MEFs or in rBs, the cell-type specific localization of HMGN binding sites coincides with the cell-type specific location of these histone marks. In sum, the cell-type-specific location of HMGN coincides with the cell-type-specific location of H3K27ac, H3K4me1 and H3K4me3 histone modifications.

Genomes contain regions in which multiple enhancers are clustered. These "super-enhancer" regions, which can be identified by their relative high density of H3K27ac, show cell-type specific location and play an important role in maintaining cell identity[4,26–28]. We find significant enrichment of HMGN occupancy in 83%, 99% and 99% of the ESCs, MEFs and rBs super-enhancers, whose location has been previously mapped[4].

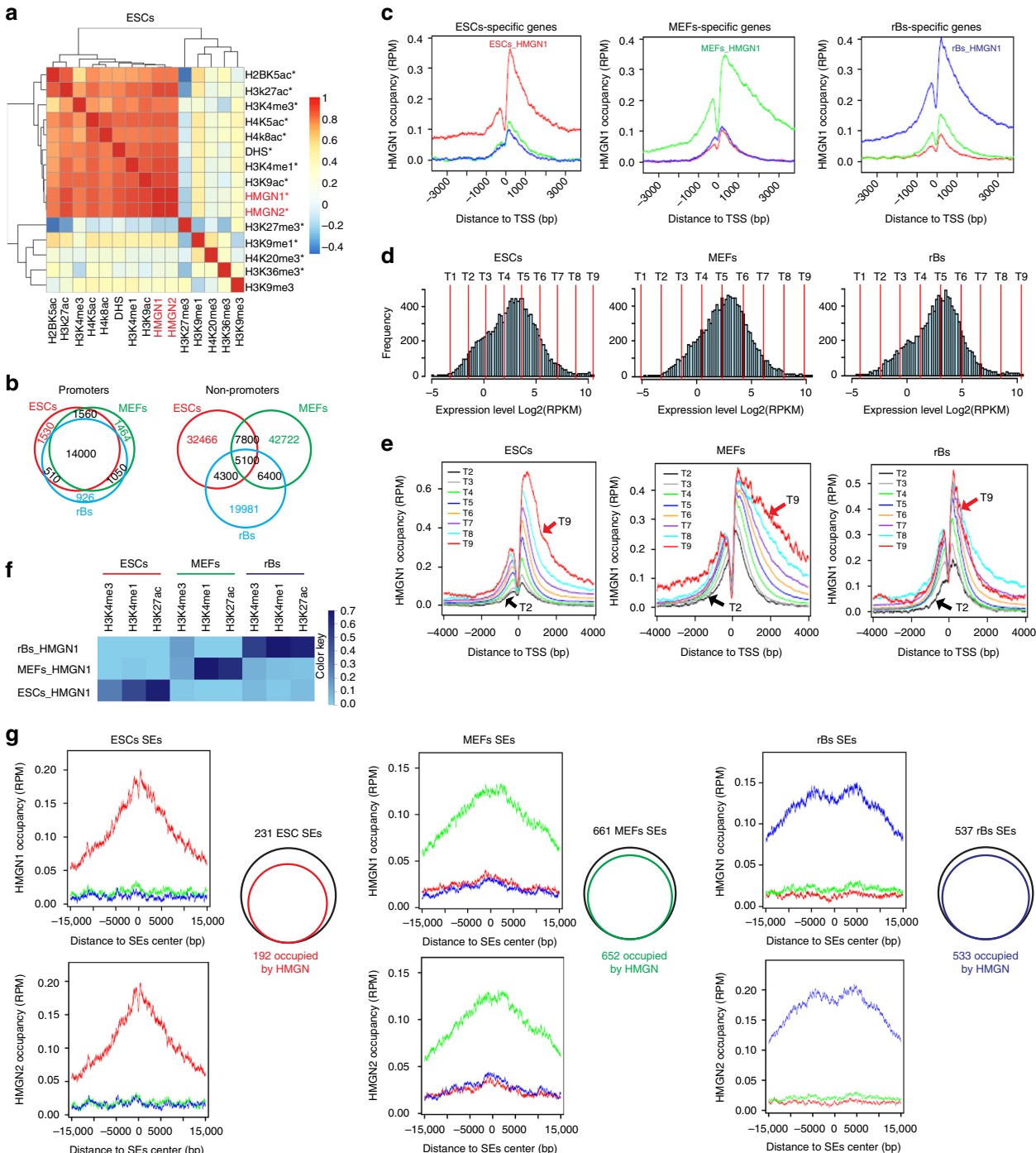

**Fig. 1** HMGNs localize to tissue specific chromatin regulatory sites. **a** Clustering map showing preferential localization of HMGN1 and HMGN2 with epigenetic marks of active chromatin in ESCs. *: data from our experiments. **b** Coincident HMGN occupancy at annotated promoters, but not at non-promoter regions in MEFs, rBs, and ESCs. **c** High HMGN1 occupancy at cell-specific expressed genes. **d**, **e** HMGN occupancy correlates directly with cell-specific gene expression levels. Genes are sorted into 9 tiers based on expression value (**d**). HMGN1 occupancy at the promoter of each group sorted by expression levels (**e**). **f** Cell-type-specific co-localization of HMGN1 with histone modifications marking cell-type-specific chromatin regulatory regions. **g** Normalized average intensity and Venn diagrams showing preferential localization of HMGN1 and HMGN2 at cell specific super-enhancers. All data from at least two biological replicates for each HMGN variant

Significantly, the HMGN occupancy at these super-enhancers is distinctly cell-type-specific (Fig. 1g and Supplementary Fig. 2b–f). Thus, in ESCs, HMGNs are enriched at ESCs-specific super enhancers but not at the MEF or rB super-enhancers, while in MEFs or in rBs, HMGNs are highly enriched at the genomic regions containing MEF- or rB-specific super-enhancers, respectively.

In sum, HMGNs show cell-type specific colocalization with epigenetic marks that identify regulatory sites of active chromatin.

**HMGNs depletion enhances somatic cell reprogramming**. The enrichment of HMGNs at cell-type-specific chromatin regulatory sites prompted us to test whether they play a role in maintaining

cell identity. We transfected MEFs derived from either WT, $Hmgn1^{-/-}$, $Hmgn2^{-/-}$, or $Hmgn1^{-/-}/Hmgn2^{-/-}$ double knockout (DKO) mice with doxycycline inducible OSKM expression vectors (Supplementary Fig. 3b) and used alkaline phosphatase (ALP) staining to evaluate the reprogramming efficiency[29]. During reprogramming, the ALP staining in either $Hmgn1^{-/-}$ or $Hmgn2^{-/-}$ MEFs shows a stronger signal than that in WT cells, but the strongest signal is observed in DKO MEFs, lacking both HMGNs (Supplementary Fig. 3a), a finding consistent with functional redundancy between HMGN variants[22]. Therefore, all subsequent experiments were performed with WT and DKO cells. Control experiments, using quantitative PCR of the *E2A-Oct4* region of the Tet-FUW-OSKM vector, verified equal vector transduction and propagation in WT and DKO MEFs, while both western and immunofluorescence verified equal expression of SOX2 and OCT4 in both cell types (Supplementary Fig. 3b-d). Likewise, cell proliferation assay revealed that the WT and DKO MEFs propagate at the same rate in either the absence or presence of Dox-induced OSKM expression (Supplementary Fig. 3e).

In repeated reprogramming experiments with OSKM expressing cells, DKO MEFs invariably showed stronger ALP signal and increased number of iPSCs colonies than WT cells (Fig. 2a, b). Likewise, in fluorescence-activated cell sorting (FACS) analyses, the DKO cells show a 4-fold higher number of cells co-expressing the pluripotency markers SSEA1 and EpCAM[30](Fig. 2c). We confirmed that the effects are indeed due to loss of HMGNs since si-RNA mediated downregulation of HMGNs in WT MEFs enhances reprogramming, while rescue of HMGN expression in DKO MEFs inhibits reprogramming (Fig. 2d, e). In an additional test, we used MEFs carrying one copy of *Col1a1::tetOP-OSKM, R26-M2rtTA* and *Oct4-GFP*, in which reprogramming is detected by the expression of Oct4-GFP following Dox induced expression of OSKM[31]. Using these cells, we find a 3-fold increase in the percentage of Oct4-GFP expressing cells in MEFs treated with HMGN-specific siRNA, as compared to cells treated with control siRNA (Fig. 2f). Taken together, all the experiments indicate that loss of HMGNs enhances the efficiency of OSKM mediated reprogramming of MEFs into iPSCs.

RNA-seq analyses of samples prepared from cells at different stages of reprogramming show differences between WT and DKO cells (Fig. 2g). In MEFs, prior to the onset of reprogramming, and in fully reprogramed iPSCs, the transcriptional profiles of WT and DKO cells are similar indicating that HMGNs do not have major effects on the steady state transcription profiles. However, during reprogramming, loss of HMGN accelerates the silencing of MEF specific genes such as *Thy1, Cd44, Dusp4, Wnt5a* and *Mcam*, and the activation of pluripotency genes such as *Nanog, Dppa4* and *Triml1* (Fig. 2h and Supplementary Fig. 3f). We detected 98 genes whose expression is downregulated more rapidly (>2-folds with FDR < 0.05) in DKO than in WT cells, and 141 genes with consistent accelerated upregulation in the DKO MEFs (Fig. 2i, k). The genes preferentially upregulated in DKO during reprogramming are enriched in gene ontology (GO) categories of stem cell maintenance, regulation of transcription and stem cell differentiation while the downregulated genes are mostly enriched in categories involved in cell migration (Fig. 2j, l). Thus, during OSKM-mediated MEF reprogramming, loss of HMGN accelerates the gradual erasure of the somatic transcriptional network and the activation of the pluripotent transcription network.

**Enhanced rate of epigenetic changes in DKO cells**. Towards understanding the molecular events associated with the enhanced rate of reprogramming in the DKO MEFs, we first profiled the genome-wide organization of HMGNs during reprogramming of WT MEFs into iPSCs. At the onset of reprogramming, day 0, both HMGN1 and HMGN2 localize mainly to MEF-specific enhancers and very few are detected at ESCs regulatory sites (Fig. 3a-d). During reprogramming, the HMGN occupancy gradually decreases at MEF-specific enhancers (Fig. 3a, c), and increases at ESC-specific enhancers (Fig. 3b, d). In fully reprogrammed iPSCs, HMGN signals are almost absent from MEF regulatory sites and are detected mostly at ESCs specific sites (Fig. 3b, d). Similar changes are seen at the cell specific super-enhancers. At the onset of reprogramming, HMGNs are present only in MEF super-enhancers; however, during reprogramming their occupancy at MEF super-enhancers decreases and at ESCs super-enhancers increases and in fully reprogrammed cells HMGNs localize to ESCs super-enhancers (Fig. 3e-h). The organization of HMGN in ESCs and iPSCs is very similar (Supplementary Fig. 4). Thus, during reprogramming, HMGNs gradually relocate from MEFs-specific to ESCs-specific chromatin regulatory sites.

Reprogramming leads to a reduction in the accessibility of MEF-specific chromatin regulatory sites and an increase in the accessibility at ESC-specific chromatin sites, a process that can be followed by Assay of Transposase Accessible Chromatin-sequencing (ATAC-seq)[7,24,32]. We find that loss of HMGN decreases the accessibility of MEF-specific regulatory sites throughout the course of reprogramming, as shown by a significant overall reduction in the ATAC signal in the DKO cells, compared to WT cells (Fig. 3i, Supplemental Fig. 4b). The temporary increase in the ATAC sensitivity of MEF-specific enhancers in both WT and DKO cells at day 4 and day 8 compared to day 0, can be attributed to the initial engagement of the reprogramming factors with the MEF chromatin and to the delay in compaction of the MEF-specific sites[6]. Indeed, in fully reprogrammed iPSCs, the accessibility of the MEF specific sites in DKO cells is lower than in WT cells (Fig. 3i) further support that loss of HMGNs promotes compaction of MEF-specific chromatin regulatory sites. The ESCs specific chromatin sites show an increase in accessibility throughout the course of reprogramming (Fig. 3j). At day 0, these sites are significantly less accessible in DKO cells; however, during reprogramming, the ESC specific sites become temporarily more accessible in DKO cells (Fig. 3j, day 4, 8, see enlargement), suggesting that loss of HMGN enhances the rate at which these sites are established. Once the iPSCs epigenetic landscape is established in fully reprogramed cells, the ESC-specific chromatin regulatory sites are again less accessible in DKO that in WT cells (Fig. 3j, iPSCs, Supplementary Fig. 4b).

In sum, at the onset of reprogramming, in MEFs at day 0, and after reprogramming into iPSCs is complete, loss of HMGNs leads to a global reduction in chromatin accessibility at both MEF-specific and ESC-specific regulatory site. However, during reprogramming, loss of HMGN preferentially enhances the relative accessibility of ESC-specific sites, suggesting that compared to WT cells, loss of HMGN enhances the rate at which an iPSC-specific epigenetic landscape is established. Thus, ATAC analyses indicates that in both MEFs and iPSCs, the presence of HMGN promotes chromatin decompaction and enhances the stability of the epigenetic landscape.

In agreement with ATAC analyses, genome wide profiling of the enhancer histone marks H3K27ac and H3K4me1, also shows that loss of HMGNs accelerates the rate of chromatin remodeling at both MEF-specific and ESC-specific enhancers. At day 0, these histone modifications are detected at MEF-specific, but not at ESC-specific enhancers (Fig. 4a, d). The signal strength is the same in DKO and WT cells, suggesting that HMGNs do not have major effects on the steady state levels of these modifications, a

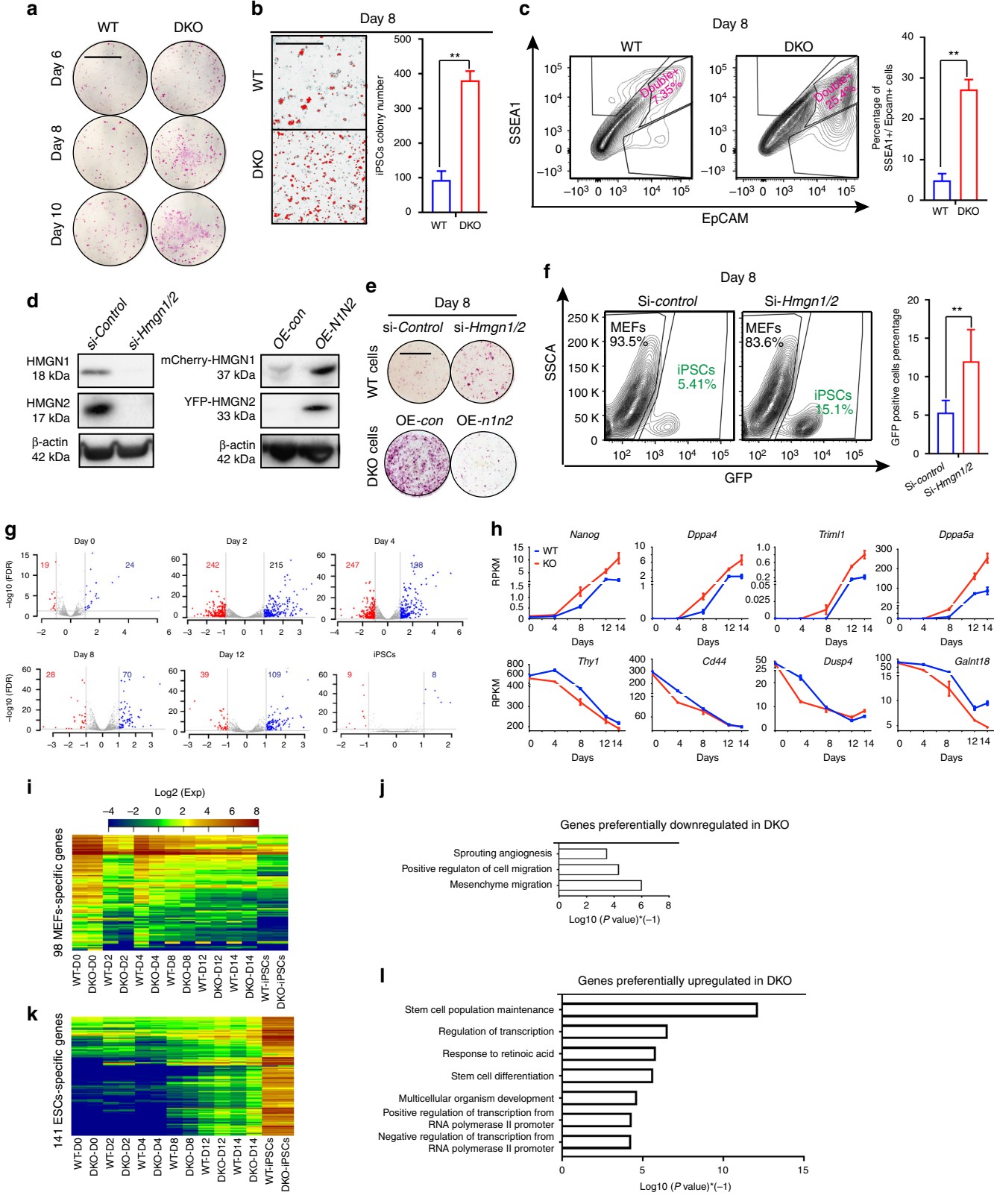

finding that is consistent with the relatively small changes in gene expression profiles (Fig. 2g). However, at reprogramming day 2, the H3K27ac signal at MEF-specific enhancers in DKO cells is only half of the signal seen in WT cells, an indication of faster erasure of MEF specific enhancer marks in DKO cells. In contrast, at day 8, the H3K27ac signal at ESCs-specific enhancers in DKO

is twice that seen in WT cells, suggesting an enhanced rate of conversion to an ESC-specific epigenetic landscape. In fully reprogramed cells, the level of modification is the same regardless whether the iPSCs were derived from WT or DKO MEFs (Fig. 4a). Similar changes are seen in the location of the H3K4me1 signals. At early reprogramming stages (Fig. 4b, day 2,

**Fig. 2** Loss of HMGNs enhances the efficiency of reprogramming MEFs into iPSCs. **a** Alkaline phosphatase staining of WT and DKO MEFs 6, 8 and 10 days, following OSKM induction. (Scale bar: 1cm) **b** Phase contrast image and number of colonies visible 8 days after OSKM induction (mean ± SD; n = 3 for each group). **c** FACS analysis of SSEA1 and EpCam expression in WT and DKO MEFs after 8 days of OSKM induction (mean ± SD; n = 4 for each group). **d** Left: western analysis of WT MEFs treated with either control, or siRNAs targeting HMGN1 and HMGN2. Right: DKO MEFs transfected with either control, or HMGN expressing vectors. **e** Alkaline phosphatase staining of MEFs 8 days after OSKM induction; top: WT MEFs treated with the indicated siRNAs; bottom: DKO MEFs treated with the indicated vectors. **f** FACS analysis of Oct4-GFP expression in MEFs carrying one copy of *Col1a1::tetOP-OSKM, R26-M2rtTA*, transfected with either control or HMGN siRNAs, 8 days after doxycycline treatment (mean ± SD; n = 2 for each group). **g** Volcano plots showing differential gene expression (fold change ≥ 1.5; FDR < 0.05) between WT and DKO MEFs following OSKM induction (data from 3 biological replicates). **h** Expression levels of representative pluripotent (*Nanog, Dppa4, Triml1, Dppa5*) and MEFs (*Thy1, Cd44, Dusp4, Galnt18*) specific genes during reprogramming of WT and DKO MEFs (data from 3 biological replicates, error bars are mean ± SD). **i, k** Heat map showing enhanced downregulation of MEF specific genes and upregulation of ESCs specific genes in DKO MEFs. **j, l** Gene Ontology categories of preferentially down regulated or upregulated in DKO MEFs during reprogramming. T-test was used in **b**, **c** and **f**, \*\*p < 0.01

4), the signal at MEF specific regulatory sites is lost faster in DKO than in WT cells, while at later stages the signal at ESC specific enhancers is higher in DKO cells than in WT cells (Fig. 4b, e). Comparable changes in the levels of these two modifications were also seen at MEF- and ESC- specific super-enhancers (Supplementary Fig. 5a, b).

At active promoters, the levels of H3K4me3 at the onset of reprogramming is high at MEF specific genes and low at ESCs specific genes in both WT and DKO cells. Loss of HMGNs had only minor effects on the levels of this modification during the early stages of reprogramming; however, at day 8, the H3K4me3 levels at ESC-specific genes in DKO was significantly higher than that of WT genes (Fig. 4c, f) suggesting a more robust activation of the ESCs specific transcription program. The dynamic changes in the epigenetic landscape of a MEF expressed gene (*Tcf19*) adjacent to an ESC expressed gene (*Pou5f1*) are visualized in the screen shot shown in Supplementary Fig. 6a. During reprogramming the reads of both HMGN1 and HMGN2 decrease at *Tcf19* locus but increase at the *Pou5f1* locus. These changes fully correlate with the changes seen in H3K27ac and H3K4me3, and with the transcription of these two genes in MEFs and in iPSCs. At non-tissue specific sites such as the *H3f3rb* gene, the levels of these modifications did not change during reprogramming (Supplementary Fig. 6b).

In sum, loss of HMGNs enhances the rate at which H3K27ac and H3K4me1 signals are lost from MEF specific enhancers and gained at ESC specific sites, and to a lesser degree the levels of H3K4me3 at cell type-specific genes. These results link the accelerated reprogramming of the DKO cells to an accelerated rate of epigenetic changes in their chromatin. Thus, the presence of HMGN reduces both the rate of OSKM-induced dynamic changes in the epigenetic landscape and the efficiency of reprogramming. We conclude that HMGNs stabilize the MEF-specific epigenetic landscape and MEF cell identity.

**DKO iPSCs maintain pluripotency potential.** Although loss of HMGN enhances the efficiency of reprogramming, the iPSCs colonies derived from WT and DKO are morphologically indistinguishable, and image analyses indicates that they express similar levels of the pluripotency markers ALP, NANOG and SSEA1 (Fig. 5a). Likewise, RNA-seq analyses indicate that the transcription profiles of the WT and DKO iPSCs including the expression levels of several pluripotent key genes such as *Sox2, Nanog, Prdm14* and *Oct4* are highly similar (Fig. 5b, c). Furthermore, WT and DKO iPSCs injected into opposite flanks of nude mice (Fig. 5d), generate teratoma that grow at comparable rates (Fig. 5e), differentiate into the three germ layers, and express similar levels of germ layer specific markers such as TUJ1 for neuroectoderm, MF20 for mesoderm, and AFP for endoderm (Fig. 5f). RNA-seq analyses of 6 WT and 6 DKO teratomas further verify that the iPSCs generated from WT and DKO MEFs are

highly similar (Fig. 5g). Altogether, these results indicate that following OSKM induced reprogramming, MEFs lacking HMGN are reprogramed into pluripotent, developmentally competent iPSCs. Thus, while HMGN affects the rate of reprogramming, they do not have marked effects on the steady state properties of the reprogrammed cells, supporting the notion that HMGNs stabilize, rather than determine cell identity.

**HMGN depletion enhances induced neuronal differentiation.** To investigate whether HMGNs safeguard cell identity, i.e., stabilize the cellular phenotype, across a different cell reprogramming system, we tested the effect of HMGNs depletion on direct conversion of MEFs into induced neurons by overexpression of the transcription factor ASCL1[33] (Fig. 6a). WT and DKO MEFs were transduced with lentivirus expressing doxycycline-inducible *Ascl1* and the neuronal differentiation efficiency evaluated by comparing the ratio of TUJ1 to ASCL1-positive cells in WT and DKO cells throughout the course of cell fate conversion. Loss of HMGN does not affect the transduction efficiency or the ASCL1 expression (Fig. 6b), but enhances the efficiency of TUJ1 expression as evaluated by quantitative microscopy (Fig. 6c-e), by western analyses (Fig. 6f) and by qPCR of the neural markers *Tubb3, Nestin* and *Map2* during neuronal induction (Fig. 6g). Thus, the presence of HMGNs reduces the rate of lineage conversion, from MEF to induced neuronal cells, supporting the notion that the presence of these proteins diminishes the rate of transcription factor induced cell fate conversion, further evidence that HMGNs stabilize cell identity.

**Discussion**
The cell-type-specific organization of enhancer elements in chromatin, which serve as binding sites for specific transcription factors, has been shown to play a key role in controlling transcription programs that maintain cell identity[4]. Our results indicate that the two major members of the HMGN nucleosome binding protein family, HMGN1 and HMGN2, bind preferentially to cell type specific enhancer elements and that their absence facilitates the ASCL1 mediated direct lineage conversion of MEFs into neurons and accelerates the OSKM induced conversion of the MEF-specific to ESCs-specific enhancer organization. Taken together, the results suggest that HMGNs stabilize cell identity by binding to cell-type-specific regulatory sites.

HMGN proteins are structural proteins, devoid of enzymatic activity and bind dynamically to chromatin without specificity for DNA sequence[15,18]. In this respect, they are different from other known regulators of cell identity such as lineage-specific transcription factors that bind to specific DNA sequence motifs, or chromatin remodelers that covalently modify chromatin components. In either MEFs or iPSCs, loss of HMGN does not noticeably alter the global organization of regulatory sites, or the

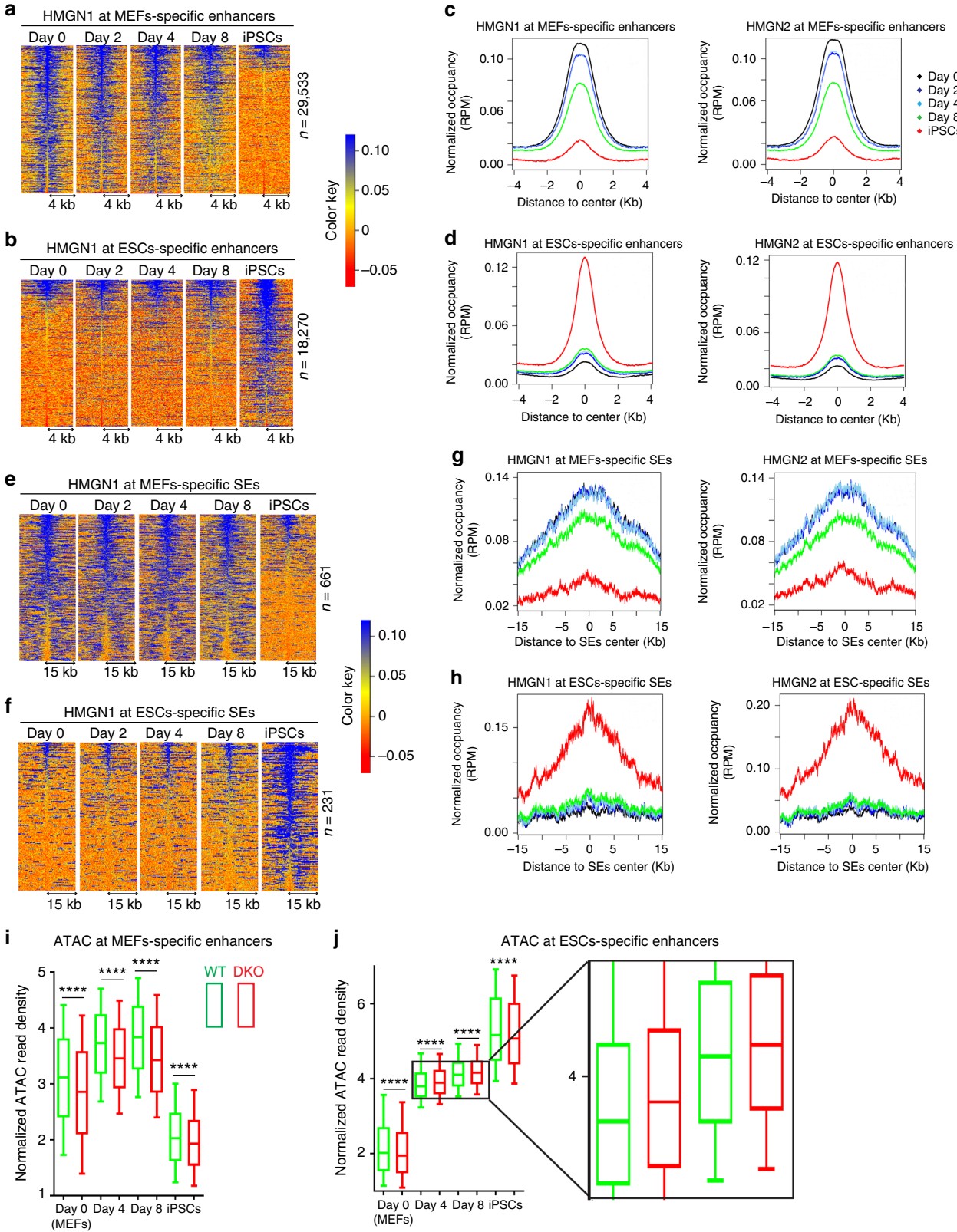

**Fig. 3** Relocation of HMGNs from MEFs specific to ESCs specific regulatory sites. Heat maps and occupancy plots of HMGN at MEFs and ESCs specific enhancers (**a**–**d**) and super-enhancers (**e**–**h**) during reprogramming. **i**, **j** Changes in chromatin accessibility during reprogramming. Box plots of normalized ATAC-seq read density at MEF- and ESC-specific regulatory sites during the reprogramming of MEFs to iPSCs. The whisker line represented the 10-90 quantiles of read density for all the regions. Significance between DKO and WT groups across all the regions was determined using a two-sided paired Wilcoxon rank-sum test. Each point is average of 3 biological replicates. ****$p < 0.0001$

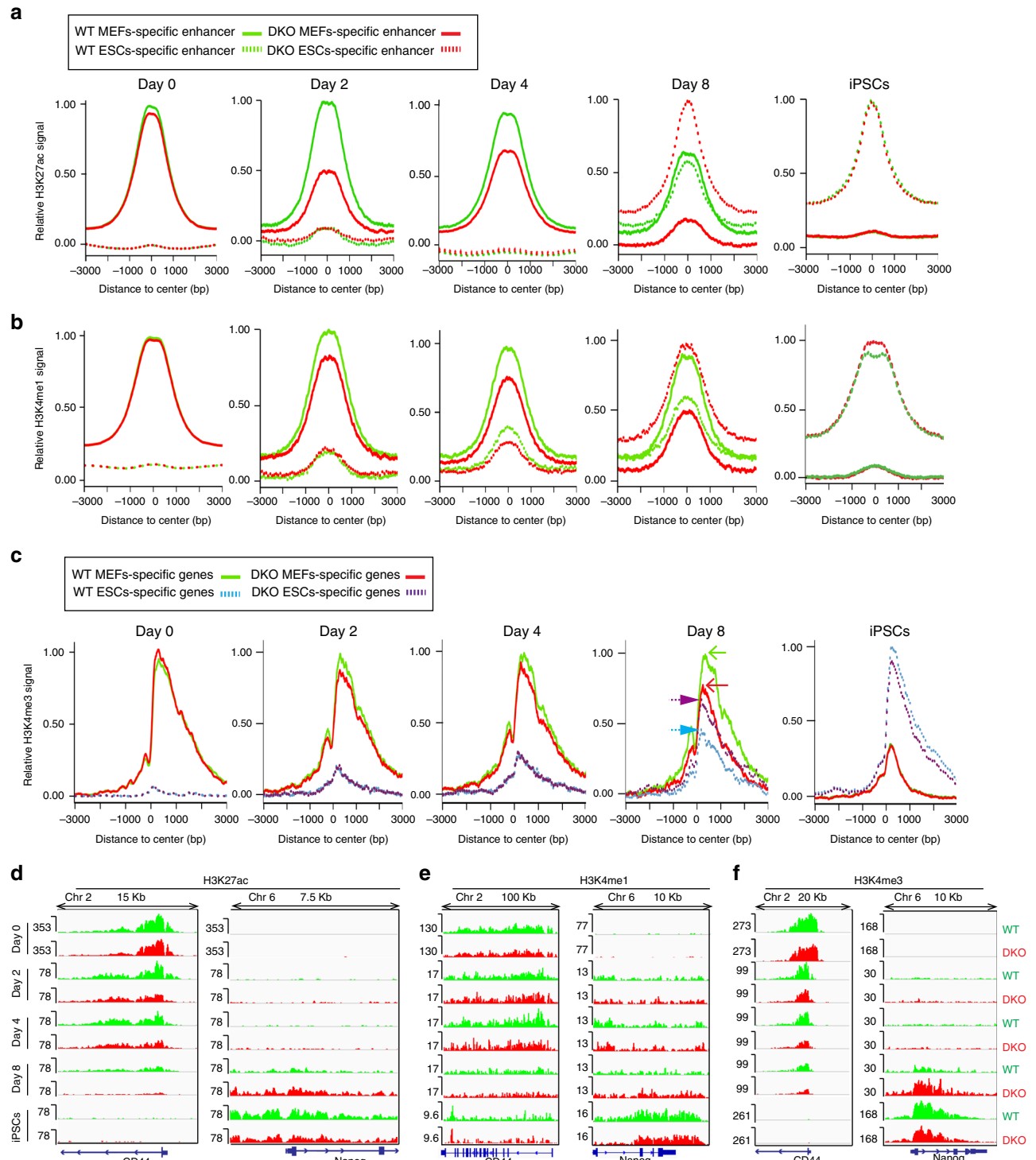

**Fig. 4** Enhanced rate of epigenetic reprogramming in DKO MEFs. **a**, **b** Occupancy plots of H3K27ac and H3K4me1 at MEF- and ESC-specific enhancers. **c**, H3K4me3 levels at promoters of genes specifically expressed in either MEFs or ESCs. **d**–**f** IGV screenshots visualizing epigenetic changes during reprogramming at a MEF-specific (*CD44*) and ESC specific (*Nanog*) expressed genes. For **a**–**c** the average of two biological replicates are plotted

level of H3K27ac and H3K4me1, two histone modifications that mark enhancer regions. However, loss of HMGNs decreases access to DNA at chromatin regulatory sites, as indicated by the decrease in ATAC signals at these sites in both MEFs and iPSCs, a finding consistent with DNase I accessibility studies in MEFs and other systems[22,34] and with numerous experiments indicating that the specific interaction of HMGN with nucleosomes decreases chromatin compaction[15,18,20]. Importantly, these effects

are global; HMGN affect the ATAC signal at numerous genomic regulatory sites rather than just a few specific sites. Taken together, the available data indicates that although HMGN binds dynamically to nucleosomes throughout the entire genome[35,36], they preferentially localize to enhancer regions and their absence decreases access to these regulatory sites.

The molecular mechanism that preferentially targets HMGN to chromatin regulatory sites is still not fully understood. HMGNs

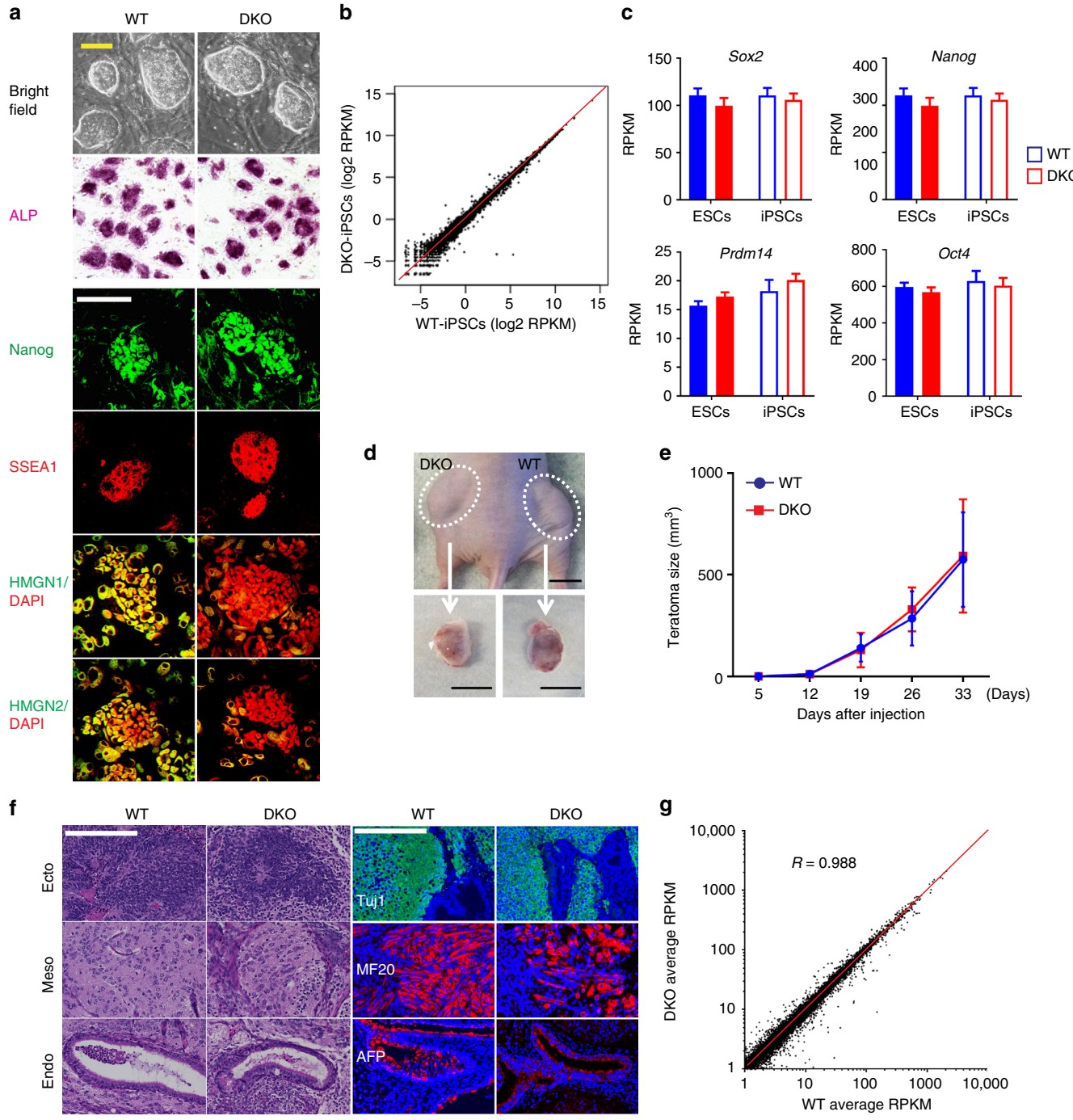

**Fig. 5** HMGN depleted iPSCs maintain pluripotency and differentiation potential. **a** Characterization of iPSCs colonies generated from WT and DKO MEFs. ALP: alkaline phosphatase assay; four bottom rows: immunofluorescence analyses of the indicated targets. DAPI visualizes DNA. Scale bar: 100 μm. **b** Scatter plot showing correlation between gene expression levels of WT and DKO iPSCs (average of 3 biological replicates). **c** Expression of select pluripotency markers in WT and DKO ESCs and in iPSCs (data was obtained from 3 biological replicates, and bar graph represents mean ± SD). **d** Teratomas derived from WT and DKO iPSCs injected into opposite flanks of nude mice. Scale bar: 1 cm. **e** Equal growth rate of teratomas derived from WT and DKO iPSCs (data was obtained from 18 replicates; error bars are mean ± SD). **f** Histological analyses of teratoma sections derived by injecting WT and DKO iPSCs into opposite flanks of nude mice. The three germ layers are identified by morphology after H&E staining (left) and by immunofluorescence with the indicated antibodies (right). Scale bar: 200 μm. **g** Scatter plot showing correlation between gene expression in teratomas derived from WT and DKO iPSCs (average of 6 biological replicates for each genotype)

bind dynamically to nucleosomes throughout the entire nucleus and potentially interact with most, or perhaps even all the nucleosomes[18,25,36]. The preferential location of HMGNs at chromatin regulatory sites detected by ChIP reflects their longer residence time at these sites. Given that HMGNs bind to nucleosomes without DNA specificity[18], the increase residence time of HMGN at these chromatin sites is likely due to the unique properties of active chromatin, such as decreased chromatin compaction, decreased H1 occupancy, and elevated acetylation of specific histone residues. Thus, HMGNs are not actively targeted

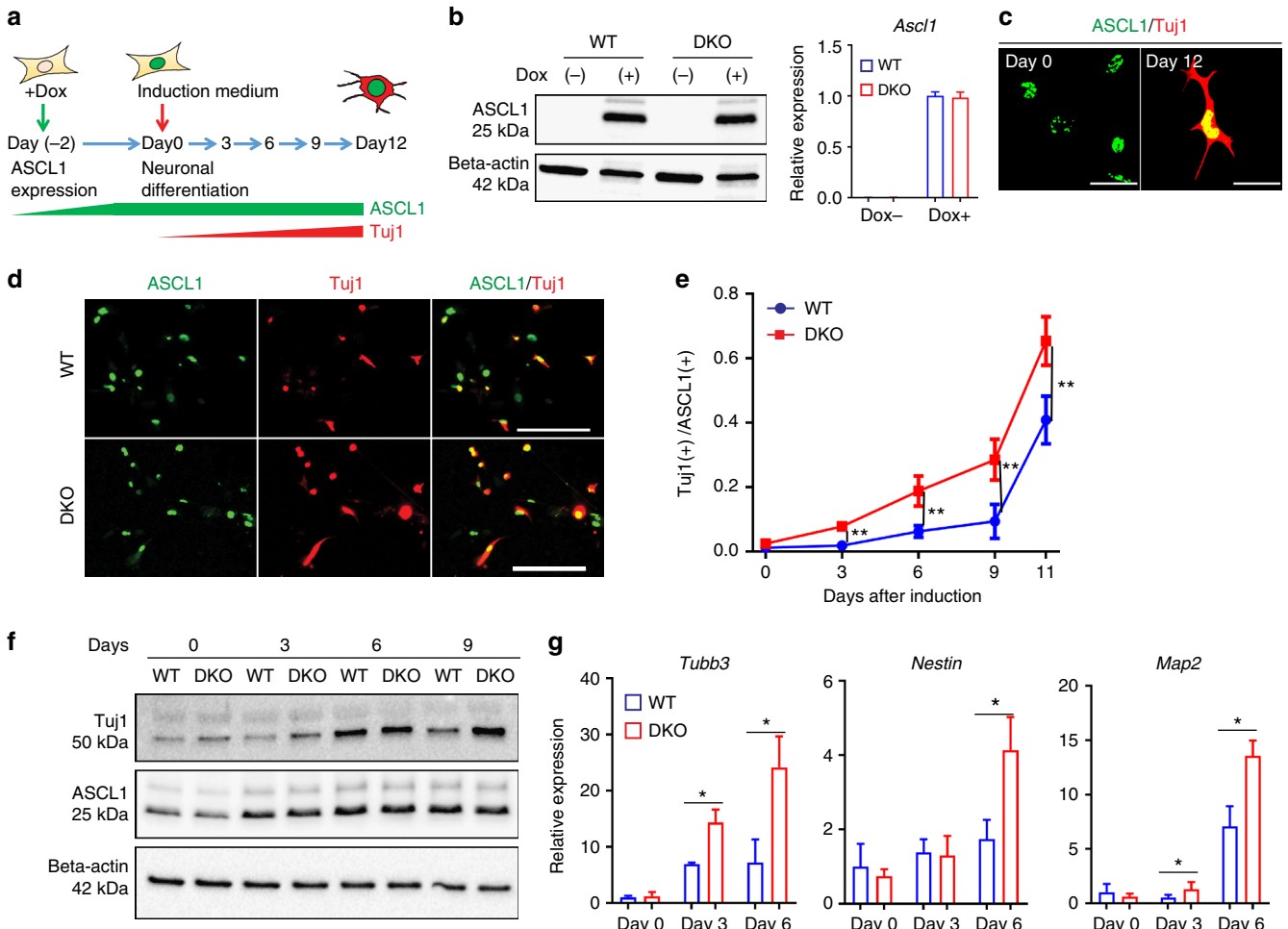

**Fig. 6** HMGN depletion enhances direction lineage conversion of MEFs to induced neurons. **a** Protocol for direct lineage conversion of MEFs to induced neurons by transcription factor ASCL1. **b** Western blot (left) and qPCR gene expression analysis (right, 3 biological replicates, scale bar: mean ± SD) showing equal expression levels of ASCL1 in WT and DKO MEFs two days after doxycycline induction. For qPCR gene expression analysis, average of three biological replicates is shown and bar graph represents mean ± SD. **c** Immunofluorescence staining of ASCL1 and TUJI in MEFs at day 0 and day 12 of doxycycline induction. Scale bar: 50μm. **d** Immunofluorescence staining of ASCL1 and TUJI in WT and DKO cells after 12 days of differentiation induction. Scale bar: 200μm. **e** Quantitative analysis of neuronal induction efficiency in WT and DKO MEFs by counting the number of TUJ1 positive cells relative to the number of ASCL1 positive cells at different induction time points (average of three biological replicates is shown, mean ± SD). **f** Western blot analysis of TUJ1 expression in WT and DKO MEFs at different induction time points. **g** Enhanced expression levels of neuron-specific markers, *Tubb3*, *Nestin* and *Map2* markers in transdifferentiated DKO MEFs. Gene expression level at each time point was normalized to *Gapdh* expression level (Average of three biological replicates are shown, mean ± SD). T-test was used in **e**, **g**. p values: *$p < 0.05$; **$p < 0.01$

to specific loci, their preferential location at enhancers reflects the unique structural properties of these sites.

Our finding that HMGNs play a role in regulating access to cell-type specific enhancers, together with numerous previous studies showing that HMGN reduce chromatin compaction[15,18,20], and oppose the action of ATP-dependent chromatin remodeling[37] provide insights into the molecular mechanisms whereby these ubiquitous proteins help safeguard cell identity. The continuous dynamic interaction of cell-type specific transcription factors with accessible chromatin regulatory sites is known to be a major factor in maintaining the integrity of the chromatin regulatory sites and the execution of cell type specific transcription programs[5,6]. A key initial step in OSKM mediated reprogramming is their binding to accessible chromatin sites and displacement of MEF specific transcription factors from MEF enhancer sites[6,7]. We suggest that the global decrease in chromatin accessibility seen in the DKO cells, reduces the dynamic interaction of cell-type specific regulators to their cognate chromatin sites thereby facilitating their displacement by the

exogenous transcription factors, whose relative robust expression is driven by the integrated lentiviral constructs. Likewise, since the inhibitory effects of HMGN on ATP dependent chromatin remodeling[37] would stabilize nucleosome position and the cell type specific epigenetic landscape, loss of HMGN would enhance the ability of chromatin remodelers to reposition nucleosomes to establish an altered epigenetic state. At the level of any single transcription factor these effects may be small and difficult to detect; it is the cumulative effect of numerous small changes that ultimately leads to accelerated erasure of the MEF specific epigenetic landscape in DKO cells. These dynamic changes during reprogramming are seen both by ATAC assays (Fig. 3) and by following the levels of histone modifications that define enhancer regions (Fig. 4).

HMGNs affect the stability of enhancer sites but do not seem to play a significant role in the establishment of these sites since the global enhancer organization in wild type and DKO cells is very similar. In this respect, HMGNs resemble chromatin modifiers such as CAF-1 which also have been shown to

safeguard rather than establish cell identity[38]. Indeed, mice lacking both HMGNs survive although they do show a wide array of mild phenotypes[22]. Gene expression analyses of cells derived from these mice show cell type-specific changes in transcription suggesting that they modulate the action of tissue specific transcription factors[39] and do not serve as general transcription factors that establish new gene expression profiles. Mice lacking HMGN1 show an impaired response to various cellular stress including DNA damage, an effect that was linked to altered binding of repair factors to chromatin[40,41]. In addition, $Hmgn1^{-/-}$ mice show a higher spontaneous incidence of certain cancers[42], further suggesting a role for HMGNs in stabilizing the identity of differentiated cells. Together, these findings support the general notion that HMGNs affect chromatin functions by modulating access of regulatory factors to their target binding sites.

The OSKM conversion of MEFs into pluripotent cells is known to yield developmentally competent iPS cells that produce chimera, differentiate into cell lineages and generate germline offspring[5,6,38]. The iPSCs produced from cells lacking HMGN are very similar to cells produces from WT mice, by several criteria. Following OSKM induction, in both WT and DKO the expression of pluripotency markers such as alkaline phosphatase, Nanog, Dppa4 and Oct 4 is upregulated while that of MEF specific genes such as Thy1 and CD44 are gradually downregulated. Significantly, the transcription profile of WT and DKO iPSCs are very similar and upon injection into nude mice the teratomas generated by the DKO iPSCs generate all three germ layers, grow at the same rate and have the same characteristics as those generated by the WT iPSCs. Taken together with previous findings that DKO mice are viable[22], the available data suggest that silencing of HMGNs does not compromise the potential of the iPSCs to differentiate into all cell lineages and generate viable offspring. However, in this study we did not examine whether the iPSCs generated from either the WT or DKO iPSCs can indeed produce chimera mice; in the absence of this assay the iPSCs identity is still not rigorously established. Nevertheless, since $Hmgn$ siRNA treatment enhances the efficiency of OSKM mediated reprogramming, modulating HMGN levels could be a means to enhance the efficiency of cell fate conversion.

Our study highlights the role of a ubiquitous family of chromatin binding proteins in maintaining a chromatin organization that optimizes the maintenance of a dynamic epigenetic landscape that can adequately respond to biological cues. HMGNs function within a dynamic network of chromatin binding protein that includes all the H1 variants which continuously alters the local chromatin organization[19,43,44]. Taken together with previous studies on the role of H1 in cell fate decisions[12,13], this study raises the possibility that the interplay between these structural proteins is part of the mechanism that maintains chromatin poised to adequately respond to stimuli that affect cell identity.

## Methods

**Mice.** $Hmgn1^{+/+}N2^{+/+}$ (WT), $Hmgn1^{-/-}$, $Hmgn2^{-/-}$, and $Hmgn1^{-/-}$; $Hmgn2^{-/-}$ (DKO) were generated in our laboratory[22]. Oct4-GFP transgenic mice (Stock 008214) and R26$^{rtTA}$; Col1a1[42] (Stock 011004) mice were purchased from Jackson lab. Heterozygous Oct4-GFP; R26$^{rtTA}$; Col1a1$^{4F2A}$, mice were generated by mating homozygous Oct4-GFP transgenic mice with R26$^{rtTA}$; Col1a1$^{4F2A}$ mice[31]. Athymic nude mice (8 week old males) were obtained from Charles River (MA). All animal procedures were done according to ACLU approved protocols as described in the NIH Guide for Care and Use of Laboratory Animals. No blinding experiments or randomization of animals was used.

**Cell culture.** Lenti-X™ 293T cell line (Clontech, Cat# 632180) was maintained in DMEM medium plus10% FBS and 1% Pen Strep. Mouse embryonic fibroblasts (MEFs) from various mice strains were prepared from E13.5 embryos and maintained in DMEM medium plus 10% FBS and 1% Pen Strep. MEFs used for reprogramming were within passage three. For reprogramming, MEFs were cultured in reprogramming medium (DMEM with 15% FBS, 1% Pen Strep, Glutamax, Sodium Pyruvate, MEM Non-Essential Amino Acids, 0.1mM β-mercaptoethanol (Thermo Fisher Scientific), 1000 U ml$^{-1}$ LIF (Millipore Sigma, Cat# ESG1106), and 2μg ml doxycycline). Mouse embryonic stem cells (ESCs) were prepared from mouse blastocysts at embryonic day 3.5 (E3.5)[45]. Mouse ESCs and iPSCs were cultured in reprogramming medium and grown on MEF feeder cells. For feeder-free culture, mouse ESCs and iPSCs were cultured in Knockout DMEM medium plus 20% KOSR (Thermo Fisher Scientific, Cat# 10828028), 1% Pen Strep, Glutamax, Sodium Pyruvate, MEM Non-Essential Amino Acids 0.1mM, β-mercaptoethanol, 1000 U ml$^{-1}$, 3 μM CHIR99021 (Selleckchem, Cat# S1263) and 1μM of PD0325901 (Selleckchem, Cat# S1036). To induce MEFs into neurons, MEFs within three passages were infected with Ascl1 lentivirus construct and cultured in N3 medium for neuronal differentiation[33]. Cells were free of mycoplasma.

**Lentivirus production.** Plasmid DNA of Tet-Fuw-OSKM (Addgene #20231), Tet-Fuw-Ascl1 (Addgene #27150), Tet-Fuw-M2rtTA (Addgene #20342), pMD2.G (Addgene #12259), and psPAX2 (Addgene #12260) were used for packaging lentivirus. Lenti-X™ 293T Cells (Clontech, Cat# 632180) were cultured in 10-cm petri dish with DMEM medium (Thermo Fisher SCIENTIFIC) plus 10% FBS and 1% Pen Strep (Thermo Fisher Scientific) at 37 °C with 5% CO$_2$. For lentivirus production, 10 μg of Tet-Fuw-cDNA (OSKM, M2rtTA or Ascl1) with 2.5 μg of pMD2.G, 7.5 μg of psPAX2 were transfected into Lenti-X™ 293T cells by using Lipofectamine 3000 transfection reagent (Thermo Fisher Scientific, Cat# L3000-015). Nine hours after initial transfection, cell culture medium was changed into DMEM medium supplemented with 5% heated-inactivated FBS. Pseudoviurs-containing culture medium was collected into sterile capped tubes 48 and 72 h post transfection, centrifuged at 500g for 10 min and filtered through 0.45 μm polyethersulfone (PES) filter membranes. The lentivirus medium was stored at -80 °C.

**Plasmid DNA and siRNAs transfection.** Plasmids DNA were transfected into cells using Lipofectamine 3000 reagent (Thermo Fisher Scientific) according to kit's protocol. SiRNAs transfections were done using Lipofectamine RNAiMAX Transfection Reagent (Thermo Fisher Scientific, Cat# 13778-075).

**MEFs reprogramming.** WT and DKO MEFs within passage 3 were used for reprogramming. Mouse MEFs were maintained in MEF medium and infected with lentivirus mixture (Tet-Fuw-OSKM and Tet-fuw-M2rtTA) with 10 μg ml$^{-1}$ polyene. The infected MEFs were trypsinized next day and re-plated on the 0.2% gelatin coated dishes with mouse MEF reprogramming medium and 2 μg ml$^{-1}$ doxycycline (Clontech, REF631311). Mouse iPSCs colonies were picked up two weeks after reprogramming.

**Cell proliferation assay.** Cell proliferation assay was performed by using CCK-8 kit according to manufacturer's instructions (Dojindo). WT and DKO MEFs at different reprogramming time points were cultured in 48-well plate and treated with 20μl CCK-8 solution, followed by one-hour incubation at 37 °C and measuring the absorbance at 450 nm using a microplate reader.

**Teratoma assay.** For teratoma formation assay[29], $5 \times 10^6$ WT or DKO iPSCs were suspended in 100 μl of PBS and subcutaneously injected into the dorsal flank of anesthetized nude mice. Three weeks after injection, the mice were euthanized and the teratomas dissected, fixed in 4% PFA overnight, and embedded into paraffin blocks. Sections were processed for histology analysis and immunofluorescence staining of three germ layer makers. To prepare RNA, 100 mg of dissected tissues were homogenized in Trizol reagent (Invitrogen) and RNA were prepared according to manufacturer's instruction. RNA was purified by RNeasy Mini Kit (QIAGEN, Cat#74104) and RNase-Free DNase Set (QIAGEN, Cat#79254).

**Phenotypic characterization of iPS cells.** Alkaline phosphatase activity (ALP) was determined using the Alkaline Phosphatase Detection Kit (MILLPORE SIGMA, Cat# SCR004). Immunofluorescence staining of iPSCs colonies was done with antibodies against Nanog (61419, Active motif), SSEA1(sc-21072, Santa-Cruz) and Oct4 (ab19857, abcam); all diluted 1:500 in antibody dilution buffer (1% BSA and 0.3% Triton X-100 in 1X PBS).

**RNA preparation and q-PCR assay of gene expression.** RNA was prepared from all cell types using RNeasy Mini kit (Qiagen) followed by on-column DNase I treatment. For RNA-seq and q-PCR gene expression analysis, 5 x iScriptTM RT super mixes (BIO-RAD) was used to reverse transcribe the RNA into cDNA. Real-time PCR amplifications were performed with the Evergreen superfast 2x PCR master mix (NewLife) and the 7900HT fast Real-Time PCR System (ABI). All real-time PCRs were performed for 40 cycles, at 95 °C for 15 s, and 60 °C for 1 min The relative expression of genes was normalized to the housekeeping gene $gapdh$. All qPCR primers and si-RNA sequences are listed in supplementary Table 1 and Table 2, respectively.

**Western blot and immunofluorescence**. For western blots, cells were lysed by CelLyticTM M buffer (Sigma) and sedimented at $16,000 \times g$ for 10 min in a Sorvall Biofuge Fresco Refrigerated Centrifuge, and the protein concentration was quantified by the Micro-BCA assay reagent (Pierce). A 10 µg protein from each sample was loaded into 4–12% SDS-polyacrylamides (Life Technologies) for electrophoresis and transferred onto PVDF membrane using iBlot Gel Transfer Stacks (Life Technologies). The following antibodies were used for Western blot experiments: Anti-Sox2 (R&D, Cat# AF2018), Anti-Oct4 (Abcam, Cat# ab19857), Anti-Ascl1 (abcam, Cat# Ab74065), Anti-Tuj1 (Santa-Cruz, Cat# sc-58888), Anti-β-actin (Sigma, Cat#A2228). For Western blot, all the primary antibodies were diluted at 1:2000 in 5% non-fat milk except that β-actin antibody was diluted at 1:20,000. The membrane was blocked by 5% non-fat milk (Biorad) and immunoblotted with antibodies by standard protocols. All the uncropped images of Western blot results related to Fig. 2d, Fig. 6b, f and Supplementary Fig. 3b are shown in Supplementary Fig. 7. Paraffin sections from teratoma were deparaffinized by xylene solutions and rehydrated in serial ethanol solutions from high to low concentration. Antigen retrieval was done by heating sections in 10 mM citric acid buffer (pH 6.0) in a microwave oven for 10 min and followed by blocking buffer (5% normal goat serum and 0.3% Triton X-100 in 1X PBS) treatment one hour at room temperature. The primary antibodies for three germ layer makers and the secondary antibodies, Anti-Tuj1(Santa-Cruz, sc-58888), Anti-MEF20 (DSHB, AB2147781), Anti-AFP (Santa-Cruz, sc-8108) were diluted into antibody dilution buffer (1% BSA and 0.3% Triton X-100 in 1X PBS). Tuj1 and AFP antibodies were diluted at 1:500, and MF20 antibody was diluted at 1:200. Nuclear staining was performed with Hochest dye (Thermo Fisher Scientific). Cultured cells in 8 well chamber slide (Millipore) were fixed by 4% PFA at room temperature for 5 min, followed by one-hour blocking buffer treatment, primary and secondary antibodies reactions. All the immunofluorescence images were taken by LSM 710 confocal microscope (Zeiss).

**Flow cytometry**. Cells were removed with enzymatic digestion and single cell suspension were stained with antibodies specific for SSEA1 conjugated to PerCP-Cy (BD PharmingenTM, Cat# 561560) and EpCam conjugated to PE (Biolegend, Cat# 118206). Oct4-GFP-expressing MEFs were directly analyzed for GFP fluorescence. Flow cytometry analyses were performed on a FACSCanto II analyzer (BD Biosciences). Data were analyzed in FlowJo (TreeStar, Inc) with gating single cells on light scatter followed by analysis of GFP or SSEA1 and EpCAM. Isotype-specific control antibodies, Rat IgG2a, k Isotype Ctrl (BioLegend, Cat#400511) and Mouse IgM, k isotype Ctrl (BioLegend, Cat# 401623) were used to establish background fluorescence.

**Transdifferentiation**. To induce WT and DKO MEFs into neurons, MEFs within the 3 passages were infected by Tet-Fuw-Ascl1 and Tet-Fuw-M2rtTA lentivirus constructs with 10 µg ml$^{-1}$ polyene[46]. Twenty-fourhours after infection, MEFs were replated at $4 \times 10^5$ cells per 35mm dish with MEF media (DMEM supplemented with 10% FBS) containing doxycycline (2 µg ml$^{-1}$) for 2 days. Then media was replaced to differentiation media (DMEM/F12 supplemented with N2 Supplement (ThermoFisher SCIENTIFIC17502-048), B27 Supplement (Thermo Fisher SCIENTIFIC, 17054-044), and Insulin-Transferrin-Selenium (Thermo Fisher SCIENTIFIC, 41400-045) containing doxycycline up to 12 days by changing the fresh media every 2 days. Ascl1 induction and neuronal differentiation was assessed by immunofluorescence staining, western blotting and qRT-PCR. For immunofluorescence staining, cells were fixed with 10% buffered formalin and permeabilized with EtOH stained with antibodies Ascl1 and Tuj1. Images were collected by using BZX-710 All In One microscopy (Keyence) and the number of Ascl1- and Tuj1-positive cells were counted by Hybrid Cell Count (Keyence). Proteins were extracted by 2x Laemmli Sample Buffer (Bio-Rad) and RNA were prepared by Trizol reagent (Invitrogen) and purified by RNeasy Mini Kit (QIAGEN) and RNase-Free DNase Set (QIAGEN).

**RNA-seq**. Transcriptome analysis were performed in WT and DKO iPSCs, WT and DKO ESCs, WT and DKO MEFs, WT and DKO reprogramming intermediate at different time point. Total RNA was isolated by RNeasy Mini Kit (Qiagen) followed on-column DNase I treatment. mRNA-seq libraries were prepared from 1 µg total RNA using the Illumina TruSeq RNA Sample Preparation Kit, following the manufacturer's instructions. Libraries were sequenced on Illumina HiSeq 2500 with read length of 125bp PE. All RNA-seq data shows average of at least 3 biological replicates.

**ChIP-seq**. All the ChIP experiments were done by using the ChIP-IT High Sensitivity kit (Active Motif, CatNo.53040). For HMGN ChIP all cells were fixed by 1% formaldehyde at 37$^0$ for 3 min; for all other ChIP, cells were fixed by 1% formaldehyde at room temperature for 10 min Chromatin was sonicated to 200–300 bp with 30 s on/ 30 s off in Bioruptor. The following antibodies were used for ChIP reactions: Anti-H3K4me1(ab8895), Anti-H3K27ac (ab4729), Anti-H3K4me3 (ab8580), Anti-H3K9me3 (ab8898), Anti-H2BK5ac(ab40866), Anti-H4K5ac (ab51997), Anti-H4K8ac (ab15823), Anti-H3K9ac (ab4441), Anti-H3K27me3 (ab6002), Anti-H3K9me1 (ab8896), Anti-H4K20me3 (ab9053), Anti-H3K36me3 (ab9050), Anti-HMGN1 (Bustin lab) and Anti-HMGN2 (Bustin lab and Cell Signaling abD9B9). 25-30 µg chromatin and 5 µg antibody were used for each ChIP

reaction. Sequencing libraries were prepared using Illumina TruSeq ChIP Sample Prep Kit (IP-202-1012) and AMPure XP beads (BECHMAN COULTER, A63881) according to manufacturer's instruction. All the library samples were sequenced on Illumina Hiseq SR 75bp. All ChIP data show average of at least two biological replicates.

**ATAC-seq**. ATAC-seq was performed as in the published protocol[47]. To analyze the chromatin accessibility in the WT and DKO iPSCs, WT and DKO reprogramming intermediate at different reprogramming stages, 50,000 cells were used for each single reaction. ATAC-seq libraries were prepared using Nextera DNA Library Preparation Kit (illumina, FC-121-1030). Briefly, cells were washed in 100 µl cold PBS and resuspended in 50 µl lysis buffer (10 mM Tris·Cl pH 7.4, 10 mM NaCl, 3 mM MgCl2, 0.1% (v/v) Igepal CA-630) for 10 min The nuclei suspension was centrifuged for for 10 min at $500 \times g$, 4 °C, followed by transposition reactions at 37 °C for 30 min (25 µl TD, 2.5 µl TDE1, 22.5 µl nuclease-free H2O). DNA was isolated using the MiniElute Kit (Qiagen, Cat# 28004) and PCR-amiplifed using the barcoded Nextera primers. Library quality control was performed using bioanalyzer. All the ATAC-seq libraries were sequenced on the Hi-Seq 2500 PE125 platform. All ATAC analyses are average of 3 biological replicates.

**Quantification and statistical analysis**. Hierarchical analysis of HMGNs ChIP data with histone modifications marks, DHS and ATAC-seq data were done in MEFs, ESC and rBs. We took 2 kb windows surrounding annotated TSS and calculated read coverages for each experiment in each window[48,49]. For both publicly available and our own datasets, including HMGN1/2 and histone marks. We then computed Spearman's rank correlation coefficient, across windows for all possible pairs of experiments and performed hierarchical clustering in each cell type. Data of H3K9me3 ChIP-seq in ESCs, and H3, H3K27me3, H3K36me3, H3k9ac, H3K4me2, h3k79me2 and H3.3 ChIP-seq in MEFs were downloaded from GSE90895. ChIP-seq data H3K27me1/2/3, H3K9me1/2/3 and ATAC-seq in rBs were downloaded from GSE82144.

**RNA-seq analysis**. RNA-seq reads were mapped to mm9 mouse reference genome using Tophat2[50] with default parameters except -N 3 (default 2) and --read-edit-dist 3 (default 2), due to our long RNA-seq read length 125 bases. The output bam file from Tophat2 were converted to read count files using htseq-count[51]. The downstream analysis was conducted with R Package EdgeR[52], Genes with expression level below 1 RPKM in all samples were removed from further analysis. Based on our RNA-seq data from three cell types: ESC, MEF and resting B cell, we defined ES specific genes as the following: Exp_ES > Exp_rB*10 and Exp_ES > 10*Exp_MEF and Exp_ES > 20 and Exp_rB < 1 and Exp_MEF < 1. The same criteria were used to define MEF and rB specific genes. For heatmaps in Fig. 2, genes induced in iPS were defined as genes whose expression level in iPS is at least four folds higher than that in any other time points. Genes down regulated were defined as genes whose expression in MEF cell (day0) is higher than that in any other time points. For Fig. 2h and S2F the analysis was performed using the RNA-seq pipeline in Partek Genome Suite. A gene was considered expressed if reads per kilobase of transcript model per million mapped reads (RPKM) was ≥ 1.0. |Fold Change| ≥ 1.5 with P value < 0.05 and fold-change > 1.5—in all cases.

**ChIP-Seq analysis**. ChIP-seq reads were aligned to mm9 mouse reference genome using Bowtie[53]. Up to 2 mismatches was allowed for each aligned read. Only uniquely aligned reads were collected for further analysis. Binding regions were identified using SICER[54] with the following parameters: effective genome size, 0.787 (78.7% of the mouse genome is mappable); window size, 100 bp for HMGN ChIP-Seq and 50 bp for histone marker ChIP-seq; gap size, 100 bp for HMGN ChIP-Seq and 50 bp for histone marker ChIP-seq. Calculation of coverage and identification of overlapping binding regions were performed with the chipseq and GenomicRanges packages in BioConductor[55]. For normalization, the calculation of coverage at any regions and the comparisons between different data sets were preceded by library size normalization. Control subtraction was carried out in the following way: coverage (exp)/N1 − coverage (control)/N2, in which exp is the data set (in.bam format) to be examined, N1 is the library size of the experimental data (exp), and N2 is the library size of the control. In this study, input sequences (DNA sequences after sonication only without immunoprecipitation) were used as a control. The function coverage that calculates genome coverage from bam files is from the chipseq package in BioConductor[55].

**Definition of super enhancers**. The coordinates of super enhancers in ES and MEF cells are downloaded from dbSUPER[56]. Super enhancers in resting B cells are obtained with program ROSE[28,57] and from our H3K27Ac ChIP-seq data.

## Data availability

The RNA-seq, ChIP-seq, and ATAC-seq data reported in this paper are available with the accession numbers: BioProject: PRJNA481982 and SRA: SRP154652. All other relevant data supporting the key findings of this study are available within the article and its Supplementary Information files or from the corresponding author upon reasonable request.

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

## Acknowledgements

This work was supported by the Center for Cancer Research, Intramural Research Program of the National Cancer Institute, NIH and by the Intramural Research Program of National Center for Biotechnology Information, National Library of Medicine, NIH.

## Author contributions

B.H., T.D. and M.B. conceived and designed the study. B.H., T.D., T.F., S.Z. and Y.P. performed experiments and analyzed the data, I.Z., D.L., W.T. and S.A. performed bioinformatic analyses, C.C.L. and F.L. performed flow cytometry analyses, B.H. and M. B. wrote the manuscript, M.B. coordinated the project. All authors approved the final version of the manuscript.

## Additional information

**Competing interests:** The authors declare no competing interests.

