## [Peer Review File · Nature Communications]

Reviewer #1 (Remarks to the Author):

This is a very interesting manuscript identifying the binding pattern and functional influence of constitutive expressed chromatin binding proteins ,HMGN1 and 2, on somatic and mouse pluripotent stem cell states.

The authors show high quality CHIP-seq analysis and that they bind enhances of somatic cell identity and when knockdown is conducted, this shuts down more efficiently somatic cell identity and enhances iPSC formation. The manuscript is well written, and data is of high quality and support the conclusions made (except in one case) which i detail below. References are accurate and methods are well detailed.

I support publishing this work following addressing this point:

1) the results of changing iN cell efficiency in the last figure are not convincing and no p values are indicated at all. is the effect significant and does HMGN knockdown enhance iN transdifferentiation ? any result is fine, but their need to be statistical analysis and conclusions adjusted accordingly.

Reviewer #2 (Remarks to the Author):

In this manuscript, using induced pluripotent cells as a model, Bing and colleagues found that, the loss of ubiquitous nucleosome binding protein HMGNs can enhanced the efficiency of iPSC generation. They further found that the loss of HMGNs could accelerate the erasure of the MEF-specific epigenetic landscape and the establishment of ESC-specific chromatin landscape. Based on this observation, they proposed that, HMGN can stabilize the cell fate but not determine the cell fate, and they also used the ASCL1 induced conversion of fibroblast into neurons to support their idea. In general, these findings are of potential interest and could be useful to the field of somatic cell reprogramming, and also offer insights for the function of ubiquitous proteins in cell reprogramming. However, the quality of the data should be improved significantly, and more evidence is needed to support the authors' conclusion.

Major concern:

1 Pluripotency assay: AP straining or SSEA1/EpCAM FACS is not a suitable way to assess pluripotency or efficiency. The author should use Oct4-GFP colonies number or other more stringent standards to estimate the reprogramming efficiency (Figure 2a-2d). In addition, how about the quality of the generated iPSC colonies? Could they give rise to chimera? The ability of germ line transmission?

2 The role of HMGNs in cell identity: The authors show that HMGNs are mainly co-located with the active epigenetic modification and binding to cell specific enhancers and also observed the shift of the epigenetic landscape from somatic cell fate to pluripotent ones during iPS generation. These would be consistent with the notion that HMGNs can serve to stabilize cell identity. Two points should be further explained. First, the binding of HMGNs to cell specific enhancers may be due to the open state of the loci in chromatin, the authors should explain the relationship between the binding of cell specific enhancer and the open-close state of the chromatin state in depth. Second, if HMGNs can stabilize cell identity, they should also be required to establish iPSCs. This is contradictory to the results obtained. Furthermore, in the DKO iPSCs, how can they maintain identity?

3 Both over-expression and DKO HMGNs impact the reprogramming process, but how do these factors co-operate with other factors to be located into the cell specific enhancers and then regulate the gene expression? Could there be an alternative explanation?

4 The effect for ASCL1 induced conversion of fibroblast into neurons seems weak. The author should calculate the significant difference with strong statistics. In addition, it's better to prove the induced "Neurons" are real neurons in physiology function.

Minor concern:

1、 Figure 1a,1e, no scale bar.

2、 Figure 4d. The Data range of the ChIP-seq data for represented genes in IGV-Screen should keep same at different time point.

Reviewer #3 (Remarks to the Author):

In this comprehensive study He et al investigate the distribution of HMGNs across different cell types and then analyse the requirement for HMGNs in cell reprogramming. They conclude that HMGNs stabilise the chromatin environment which in turn modulates transcription and epigenetic marks. This appears to be a carefully undertaken study and makes an important contribution to the field.

The authors clearly show a change in reprogramming rate in absence of HMGNs. Are there any negative consequences to removing HMGNs? Presumably loss of HMGNs would allow H1 binding, making chromatin less accessible to regulatory elements. I can understand how this could affect gene silencing but why would it increase the upregulation of genes?

Title

I don't like the emphasis on "stabilising cell identity". This is used in the title and end of introduction and in many other places. Is it possible to be more precise e.g. for a title

"Binding of HMGN proteins to cell specific enhancers affects transcription and epigenetic marks". To me it is clear that HMGNs are regulating chromatin structure that in turn affects transcription factor binding and transcription, important for cell identity. HMGNs are one of the chromatin regulators, but the TFs are still the master controllers. It's not necessary to big up the role for the HMGN proteins, as is demonstrated well in the manuscript they have an important role in stabilising the chromatin environment, modulating transcription.

Abstract line 29

The authors write "...modulate the plasticity of the chromatin epigenetic landscape...". I feel accessibility is a better term than plasticity.

Introduction

Page 3, Line 58. Define epigenetic plasticity

Page 4, line 67. Instead of epigenetic marks maybe histone modifications could be used instead.

Page 4, line 69. The authors write "play a role in cell fate decisions". Presumably you mean transcription factor binding?

Results

Figure 1

The meta-analysis of HMGN binding is interesting but I would like to see HMGN binding over some specific genomic regions. Ideally in regions of the genome where there might be differences between cell types.

Page 6, line 127. Instead of epigenetic marks maybe histone modifications could be used instead.

Panel 1g on super-enhancers should be moved to a supplementary figure

Figure 2

Page 7, line 174. Maybe a better term than “epigenetic reorganization” could be found.

Page 8, line 196. The discussion about initial engagement of TFs sounds very speculative. What is the evidence? This might be better saved for the discussion.

Figure 3/4

Panel 3e,f,g,h should be moved to supplementary

I found figure 3i and 3j very unconvincing and quite difficult to explain. The differences are extremely small and I can't imagine biologically relevant. This section could be deleted with no impact on the manuscript. If the authors want to keep it they would need to show examples at specific genomic loci.

Page 10, line 250. The final sentence is vague and should be deleted.

Figure 6

Page 11, Line 271. What do the authors mean by “HMGNs safeguard cell identity”? I am sure this could be better worded.

Page 11, line 282. I think it is best to avoid a phrase like “HMGNs stabilise cell identity”. HMGNs influence chromatin structure, that in turn regulates transcription factor access, and this then alters cell identity. The results in the manuscript are sufficiently clear that it is not necessary to “hype” the role of HMGNs.

RESPONSE TO REVIEWERS.

We thank the reviewers for taking the time to review the manuscript and for their constructive remarks. Below is the answer to each of the comments by the 3 reviewers.

REVIEWER #1 (Remarks to the Author):

This is a very interesting manuscript identifying the binding pattern and functional influence of constitutive expressed chromatin binding proteins, HMGN1 and 2, on somatic and mouse pluripotent stem cell states.

The authors show high quality CHIP-seq analysis and that they bind enhances of somatic cell identity and when knockdown is conducted, this shuts down more efficiently somatic cell identity and enhances iPSC formation. The manuscript is well written, and data is of high quality and support the conclusions made (except in one case) which i detail below. References are accurate and methods are well detailed.

I support publishing this work following addressing this point:

1) the results of changing iN cell efficiency in the last figure are not convincing and no p values are indicated at all. is the effect significant and does HMGN knockdown enhance iN transdifferentiation? any result is fine, but their need to be statistical analysis and conclusions adjusted accordingly.

Answer: We have assessed reprogramming efficiency by both immunofluorescence and quantitative PCR of 3 neuronal markers. As requested, in Fig. 6 we have added p values in panels e, g and the legend. The data show statistically significant changes starting day 3 after induction of neuronal transdifferentiation.

REVIEWER #2 (Remarks to the Author):

In this manuscript, using induced pluripotent cells as a model, Bing and colleagues found that, the loss of ubiquitous nucleosome binding protein HMGNs can enhanced the efficiency of iPSC generation. They further found that the loss of HMGNs could accelerate the erasure of the MEF-specific epigenetic landscape and the establishment of ESC-specific chromatin landscape. Based on this observation, they proposed that, HMGN can stabilize the cell fate but not determine the cell fate, and they also used the ASCL1 induced conversion of fibroblast into neurons to support their idea. In general, these findings are of potential interest and could be useful to the field of somatic cell reprogramming, and also offer insights for the function of ubiquitous proteins in cell reprogramming. However, the quality of the data should be improved significantly, and more evidence is needed to support the authors' conclusion.

Major concern:

1 Pluripotency assay: AP straining or SSEA1/EpCAM FACS is not a suitable way to assess pluripotency or efficiency. The author should use Oct4-GFP colonies number or other more stringent standards to estimate the reprogramming efficiency (Figure 2a-2d). In addition, how about the quality of the generated iPSC colonies? Could they give rise to chimera? The ability of germ line transmission?

ANSWER: We assessed pluripotency not only using ALP staining and SSEA1/EpCAM FACS sorting, but also used Flow cytometry to measure Oct4-GFP expression in 8 day cultures generated by fibroblast obtained from genetically altered mice carrying *Col1a1::tetOP-OSKM*, *R26-M2rtTA* and *Oct4-GFP*, as shown in Fig 2f. As previously demonstrated by others (see refence 31 and new reference 38), in these cells, Oct4_GFP expression following Dox induced OSKM expression is a reliable measure of their

pluripotency. Significantly, we downregulated HMGN levels by treating cells with specific siRNAs and found that within 8 days of reprogramming induction, downregulation of HMGNs leads to a significant increase in the percentage of Oct 4 positive cells (Fig 2 f). In addition, the data in Fig 2h and supplementary figure 3 indicates that the DKO cells express several pluripotency markers just as wild type cells do, albeit the upregulation is faster in DKO cells. Furthermore, injection of the DKO cells into nude mice gave rise to all 3 germ layers and both the growth rate the transcription profiles of the DKO teratomas was the same as that of the WT teratomas. Thus, by several accepted criteria, the pluripotency of the DKO cells is the same as that of the WT cells, which are known to produce chimera. In the revised manuscript we have modified panels d,e, and f in Fig 2. In d,e we added the description of the transfected cells, in panel f we now indicate that the data is from day 8 cultures. In addition, we have added a new paragraph at the bottom of page 13 which emphasizes all the points mentioned here.

The major emphasis of this manuscript is the role of HMGNs in maintaining cell fate. Although our manuscript is not concerned with the technical and practical aspects of iPSCs, we certainly agree with the reviewer that the quality of the generated iPSCs are an extremely important point, especially since in principle our results may have practical applications. Our data shows that the transcription profile and morphology of iPSCs generated from WT cells are very similar to those generated from DKO MEFs (Fig 5 a-c). In addition, our studies with nude mice clearly show that by several criteria, including the generation of 3 germ layers and transcription profile, the teratomas generated from both WT and DKO iPSCs are indistinguishable (Fig 5d-g). Furthermore, we already reported that mice lacking HMGN1 and HMGN2 are viable (Reference 22). It is well documented that iPSCs generated from WT MEFs can yield chimera capable of germ line transmission. Taken together the available data strongly suggest that the DKO iPSCs will also generate chimera and there will be germ line transmission. For practical uses, tests need to be done with human cells. We thank the reviewer for raising these important questions. In the revised version we discuss these points in the new paragraph that starts at the at the bottom of page 13.

2 The role of HMGNs in cell identity: The authors show that HMGNs are mainly co-located with the active epigenetic modification and binding to cell specific enhances and also observed the shift of the epigenetic landscape from somatic cell fate to pluripotent ones during iPS generation. These would be consistent with the notion that HMGNs can serve to stabilize cell identity. Two points should be further explained. First, the binding of HMGNs to cell specific enhances may due to the open state of the loci in chromatin, the authors should explain the relationship between the binding of cell specific enhance and the open-close state of the chromatin state in depth. Second, if HMGNs can stabilize cell identity, they should also be required to establish iPSCs. This is contradictory to the results obtained. Furthermore, in the DKO iPSCs, how can they maintain identity?

ANSWER: Regarding the first point. We added a paragraph (2nd paragraph on PAGE 12) to point out that the binding of HMGNs to enhancers may be due to specific properties of active chromatin. Regarding the second point, as discussed in the manuscript, HMGNs stabilize rather than determine cell identity. Thus, the DKO cells can maintain their identity but when the identity is challenged by overexpressing OSKM or other transcription factors, they lose their identity faster than WT cells. Indeed, it has been previously shown by others that chromatin remodelers can affect the stability of cell identity. For example, in a landmark manuscript (reference 38) it has been shown that the histone chaperon CAF-1 has similar properties. Just like we found with HMGNs, MEFs lacking CAF-1 are more easily converted to

iPSCs and mice lacking CAF-1 survive. It has been demonstrated (reference 22,45, see also <http://tools.mouseclinic.de/phenomap/jsp/annotation/public/phenomap.jsf>) that mice lacking HMGNs are more susceptible to various biological stresses. Thus, as was shown with other chromatin modifiers, cells lacking HMGNs survive and maintain their identity, but their ability to maintain cell identity is weakened making them more susceptible to cell fate alterations, including cancer. The difference between stabilizing and establishing cell identity is mentioned in several places in the manuscript including the abstract. In the revised manuscript we added a sentence to the second paragraph of page 13 to further emphasize the distinction and to compare it to the effects of CAF-1.

3 Both over-expression and DKO HMGNs impact the reprogramming process. but how do these factors co-operate with other factors to be located into the cell specific enhancers and then regulate the gene expression? Could there be an alternative explanation?

ANSWER: We have not shown that overexpression of HMGN affects the reprogramming factor. What we show is that overexpression of HMGN in DKO cells rescues the wild type phenotype (Fig 2e). This is a very important control to show that phenotype seen is indeed due to loss of HMGNs rather than to some minor genotypic differences between WT and DKO cells. This is mentioned in the first paragraph of page 7.

4 The effect for ASCL1 induced conversion of fibroblast into neurons seems weak. The author should calculate the significant difference with strong statistics. In addition, it's better to prove the induced "Neurons" are real neurons in physiology function.

Minor concern:

1. Figure 1a,1e, no scale bar.

2. Figure 4d. The Data range of the ChIP-seq data for represented genes in IGV-Screen should keep same at different time point.

ANSWER: As requested, we added p-values to the relevant panels in figure 6 to show that the data is statistically significant. Indeed, for use of HMGN treatment in practical medical procedures it is imperative to show that the neurons are physiologically competent. As elaborated above, this needs to be done in human cells. Similar to the experiments done in the manuscript describing the effects of CAF1 on cell identity (reference #38), our experiments with neuronal system serve to strengthen the notion that HMGNs stabilize cell identity.

1. I think the reviewer meant Fig 2a and 2e rather than Fig 1a,1e. These images are from 12-well culture plates containing iPSCs colonies. Scale bars are not generally added to images of culture plates. Nevertheless, as requested we now added scale bars.

2. It is not possible to keep the ranges at the same level because the IGV represent raw data of histone modifications at different days of differentiation. It is well documented that the level of the histone modifications changes during differentiation. Thus, the RPKM showing the levels of modification at a certain locus changes during differentiation. Our study compares WT and DKO cells therefore the data range of WT and DKO cells are indeed kept the same thereby allowing comparison of histone modification in WT to those in DKO cells during differentiation. In the IGVs shown in figure 4d, the data range for all the modifications at all time points of WT cells are identical to those of DKO cells.

REVIEWER #3 (Remarks to the Author):

In this comprehensive study He et al investigate the distribution of HMGNs across different cell types and then analyse the requirement for HMGNs in cell reprogramming. They conclude that HMGNs stabilise the chromatin environment which in turn modulates transcription and epigenetic marks. This appears to be a carefully undertaken study and makes an important contribution to the field.

The authors clearly show a change in reprogramming rate in absence of HMGNs. Are there any negative consequences to removing HMGNs? Presumably loss of HMGNs would allow H1 binding, making chromatin less accessible to regulatory elements. I can understand how this could affect gene silencing but why would it increase the upregulation of genes?

ANSWER: Indeed there are negative consequences in removing HMGNs. We publish detail phenotypic changes in DKO mice (<http://tools.mouseclinic.de/phenomap/jsp/annotation/public/phenomap.jsf> , and reference 22,45 in this manuscript). Additional examples, from other laboratories, can be found in PubMed. The present manuscript shows that loss of HMGN decreases the ability of cells to maintain cell identity, a significant new phenotype which explains some earlier findings. Decrease (not loss) in the ability to maintain cell identity may be the underlying cause of some of the phenotypes seen in DKO cells and mice, including increase susceptibility to cancer. These points are discussed on page 13. Regarding upregulation of genes in DKO cells, it is well documented that changes in the levels of architectural proteins such as H1 and HMGs can lead to chromatin changes that both upregulate and downregulate gene expression (reference 12). A possible explanation is that downregulation of a transcriptional repressor may lead to activation of other genes.

Title

I don't like the emphasis on "stabilising cell identity". This is used in the title and end of introduction and in many other places. Is it possible to be more precise e.g. for a title "Binding of HMGN proteins to cell specific enhancers affects transcription and epigenetic marks". To me it is clear that HMGNs are regulating chromatin structure that in turn affects transcription factor binding and transcription, important for cell identity. HMGNs are one of the chromatin regulators, but the TFs are still the master controllers. It's not necessary to big up the role for the HMGN proteins, as is demonstrated well in the manuscript they have an important role in stabilizing the chromatin environment, modulating transcription.

ANSWER: We emphasized "cell identity" because it highlights an important general biological function of HMGNs. It has been previously reported that changes in HMGNs can lead to transcriptional changes. Here we emphasize a very important biological consequence of these changes. We have taken the liberty to use this title because a very similar title has been used for a similar landmark study with CAF1 that has been published 3 years ago in Nature: "The histone chaperon CAF-1 **safeguards** somatic cell identity" (reference # 38 in our manuscript).

Abstract line 29

The authors write "...modulate the plasticity of the chromatin epigenetic landscape...". I feel accessibility is a better term than plasticity.

ANSWER: we do not know whether HMGN affects are solely due to changes in accessibility. Our data

shows that HMGN modulate the rate at which the epigenetic changes occur during preprogramming.

Introduction

Page 3, Line 58. Define epigenetic plasticity

ANSWER: according to the dictionary “plasticity” is defined as: “the quality of being easily shaped or molded”. This definition describes well our finding that loss of HMGN affect the ease at which chromatin can be changed as measured by several epigenetic marks. This term has been used in the literature before (see: <https://www.ncbi.nlm.nih.gov/pmc/articles/PMC4039141/>). As requested, in the revised manuscript we added a short descriptor immediately after the word “plasticity” (see page 3 last paragraph).

Page 4, line 67. Instead of epigenetic marks maybe histone modifications could be used instead.

ANSWER: We also see differences in ATAC sensitivity and we previously reported changes in DNase I (reference #22). Previous studies from several laboratories showed that HMGN compete with H1 therefore it is possible that loss of HMGN leads to additional “epigenetic “changes.

Page 4, line 69. The authors write “play a role in cell fate decisions”. Presumably you mean transcription factor binding?

ANSWER: As elaborated above there are other factors than just transcription factor binding that play a role in cell fate decision. We mentioned CAF-1 (reference #38), H1 (reference #13), HMGA (reference #16), however there are several other chromatin modifiers that play a role in this process. In the introduction we did not want to limit ourselves to one single possible scenario.

Results

Figure 1

The meta-analysis of HMGN binding is interesting but I would like to see HMGN binding over some specific genomic regions. Ideally in regions of the genome where there might be differences between cell types.

ANSWER: Supplementary Fig 2 e,f show IGVs of cell specific HMGN binding at specific genomic loci.

Page 6, line 127. Instead of epigenetic marks maybe histone modifications could be used instead.

ANSWER: They also localize to, and affect DNase I sensitivity. We wish to emphasize that in a broad sense HMGNs can affect the “epigenetic landscape” (this term has been coined by others, it is not original to us).

Panel 1g on super-enhancers should be moved to a supplementary figure

ANSWER: This panel shows specific localization of both HMGN1 and HMGN2 to cell specific super-enhancers. It is primary data and one of the most significant new findings. We feel quite confident that this important new observation should remain in the main figure. In fact, the “instructions to authors” specifies that primary data should be in the main text and not supplement.

Page 7, line 174. Maybe a better term that “epigenetic reorganization” could be found.

ANSWER: As requested, we have changed the word to “epigenetic changes”.

Page 8, line 196. The discussion about initial engagement of TFs sound very speculative. What is the evidence? This might be better saved for the discussion.

ANSWER: The initial engagement of TFs with MEF chromatin is not a hypothesis. It is an experimental finding described (by others, not us) in reference #6. It is for this reason we mention it in the results.

Panel 3e,f,g,h should be moved to supplementary

ANSWER: These panels show relocation of HMGN from MEF specific super-enhancers to ESCs super enhancers during reprogramming. The data in this figure shows that this relocation occurs both at cell specific enhancers and at cell specific super enhancers. We feel that these are important new, primary observations which should be presented in the main figures. We would prefer to keep them in the main figure, together with the panels showing the changes at enhancers. It is an important novel observation.

I found figure 3i and 3j very unconvincing and quite difficult to explain. The differences are extremely small and I can't imagine biologically relevant. This section could be deleted with no impact on the manuscript. If the authors want to keep it they would need to show examples at specific genomic loci.

ANSWER: Indeed the differences are small but in fact they are significant. The data indicates detectable changes in thousands of ATAC sites. The box plots show bioinformatic analyses of many thousands of ATAC sites and the p value is extremely significant. Such "box plots" is a customary way to describe and analyze genome wide epigenetic changes which at single sites are small but because they occur at a very large number of sites they are statistically highly significant. Examples from the recent literature showing similar analyses and results can be found in references: PMID: 29337183 (*Fig1*) and in 26659182: (*Fig5*). As requested, specific examples are shown in Supplementary figure 4b

Page 10, line 250. The final sentence is vague and should be deleted.

ANSWER: We would prefer to keep the sentence since it gives a succinct summary of some of the conclusions.

Page 11, Line 271. What do the authors mean by "HMGNs safeguard cell identity"? I am sure this could be better worded.

ANSWER: "Safeguard cell identity" has been used in several manuscripts including the title of the landmark manuscript listed as reference # 38. It conveys adequately the role of HMGNs in stabilizing cell identity. We really would like to keep this accepted concise nomenclature rather than going through a complicated and convoluted discussion. Nevertheless, as requested by the reviewer we added a short descriptor ("i.e. stabilize the cellular phenotype"), please see page 11, first paragraph.

Page 11, line 282. I think it is best to avoid a phrase like "HMGNs stabilize cell identity". HMGNs influence chromatin structure, that in turn regulates transcription factor access, and this then alters cell identity. The results in the manuscript are sufficiently clear that it is not necessary to "hype" the role of HMGNs.

ANSWER: We do agree that any idea could be expressed in several ways. We think that "stabilize cell identity", a term that has been used by others, adequately and concisely describes the cellular function of HMGNs, as we can deduct from the available data. As mentioned above, as requested by the reviewer, we did add a short descriptor. But we would like to use this terminology rather than expanding and reiterating sentences used in numerous previous articles.

We hope that we answered adequately all the comment of the reviewers. Again, we do thank the reviewers for reviewing our manuscript and for their remarks.

Reviewer #1 (Remarks to the Author):

The authors have addressed the points raised by the reviewers in a satisfactory way.

Reviewer #2 (Remarks to the Author):

The revision addressed some of my concerns. Here are some concerns left unaddressed experimentally.

1) chimera assay for the generated cells lacking both protein. The authors may be right that given what is known, those cells should give rise to chimera. But, a chimera assay is easier than the teratoma assay shown. I strongly suggest that the authors generate chimera to show vigorously the iPSCs being authentic.

2) My concern about the binding mechanisms of HMGN1 and 2 to open chromatin remains. Can the authors mutate HMGN1 and 2 in their rescue experiments to identify the protein regions required for the observed effects?

Reviewer #3 (Remarks to the Author):

I am happy with the responses to my questions and comments

Response to reviewers.

Reviewers #1 and #3 had not comments.

Reviewer #2 requested two new experiments.

We do not agree with these requests and wrote a “letter to the editor”.

Reviewers' comments:

Reviewer #1 (Remarks to the Author):

The authors have addressed the points raised by the reviewers in a satisfactory way.

Reviewer #2 (Remarks to the Author):

The revision addressed some of my concerns. Here are some concerns left unaddressed experimentally.

1) chimera assay for the generated cells lacking both protein. The authors may be right that given what is known, those cells should give rise to chimera. But, a chimera assay is easier than the teratoma assay shown. I strongly suggest that the authors generate chimera to show vigorously the iPSCs being authentic.

2) My concern about the binding mechanisms of HMGN1 and 2 to open chromatin remains. Can the authors mutate HMGN1 and 2 in their rescue experiments to identify the protein regions required for the observed effects?

Reviewer #3 (Remarks to the Author):

I am happy with the responses to my questions and comments

Michael Bustin, PhD

Therefore, in this letter I am asking you to reconsider your request for performing the chimera assay for the following major reasons:

1. Granted, the chimera assay is a very good test for iPSCs pluripotency. However, the manuscript already contains several assays showing cell pluripotency in both Wild type and Knock-out cells, including alkaline phosphatase staining (Fig 2), expression of pluripotency marker genes, teratoma assay showing differentiation into the three germ layers (Fig. 5), and detailed RNAs-seq analyses (6 biological replicates!) showing identical transcription profiles of WT and HMGN knock-out teratomas (Fig 5). By all accepted criteria, the iPSCs generated from WT cells are indistinguishable from those generated from the HMGN knock-out cells. Furthermore, it is well documented that mice lacking HMGNs are born. Given these findings and the known ability of WT iPSCs to generate chimera mice, it would not be meaningful to test whether the knock-out iPSCs can also generate chimera mice.

Thus, showing that our iPSCs, can also generate chimera would not be meaningful and would not add any new, unexpected information.

2. Chimera is indeed a good assay for iPSCs but we do not study the generation of iPSCs. The manuscript contains two major new findings. First, that the ubiquitous HMGN protein localize to chromatin regulatory regions, including to the cell-type specific super-enhancers (Most of data in Fig 1). Second, that HMGN proteins stabilize cell identity. Loss of HMGN accelerates transcription factor mediated changes in cell identity in two separate experimental system. In other words, loss of HMGNs destabilizes the ability of MEFs to maintain their identity and facilitates the establishment of new cell identities. Neither of these two major points require experimental proof that we can generate viable chimera. We study the kinetics of epigenetic changes associated with the changes in cell fate and demonstrate that HMGNs stabilize the existing MEF-specific cell fate. We do not develop new methods of generating iPSCs, do not characterize new properties of iPSCs, or improve the therapeutic potential of such cells.

Thus, showing that the iPSCs can generate chimera would not strengthen the major findings of the manuscript.

3. The chimeras will be prepared by our animal facility, they are not generated in our laboratory. We were informed by our animal facility that obtaining approval from the NIH animal committee and then generating the chimera would take over 4 months, perhaps even longer.

Thus, generating chimera will significantly delay the publication of this manuscript, but not add significant new information or provide additional experimental support for the major findings of our manuscript.

Please note that neither reviewer 1 nor reviewer 3 felt that this is necessary. In fact, even reviewer #2 writes: “The authors may be right that given what is known, those cells should give rise to chimera. But, a chimera assay is easier than the teratoma”. Thus, this reviewer does not suspect that our findings are wrong he just would like to get more information which, as mentioned above, will not strengthen the manuscript.